# SAM-CLIP:
# Merging Vision Foundation Models towards Semantic and Spatial Understanding

## Abstract

The landscape of publicly available vision foundation models (VFMs), such as CLIP and Segment Anything Model (SAM), is expanding rapidly. VFMs are endowed with distinct capabilities stemming from their pre-training objectives. For instance, CLIP excels in semantic understanding, while SAM specializes in spatial understanding for segmentation. In this work, we introduce a simple recipe to efficiently *merge* VFMs into a unified model that assimilates their expertise. Our proposed method integrates multi-task learning, continual learning techniques, and teacher-student distillation. This strategy entails significantly less computational cost compared to traditional multi-task training from scratch. Additionally, it only demands a small fraction of the pre-training datasets that were initially used to train individual models. By applying our method to SAM and CLIP, we derive SAM-CLIP : a unified model that amalgamates the strengths of SAM and CLIP into a *single backbone*, making it apt for edge device applications. We show that SAM-CLIP learns *richer visual representations*, equipped with both localization and semantic features, suitable for a broad range of vision tasks. SAM-CLIP obtains improved performance on several head probing tasks when compared with SAM and CLIP. We further show that SAM-CLIP not only retains the foundational strengths of its precursor models but also introduces *synergistic functionalities*, most notably in zero-shot semantic segmentation, where SAM-CLIP establishes new state-of-the-art results on 5 benchmarks. It outperforms previous models that are specifically designed for this task by a large margin, including +6.8% and +5.9% mean IoU improvement on Pascal-VOC and COCO-Stuff datasets, respectively.

## 1 Introduction

Vision Foundation Models (VFM) such as CLIP (Radford et al., 2021), SAM (Kirillov et al., 2023), MAE (He et al., 2022), and DINOv2 (Oquab et al., 2023) provide strong backbones that work well for a wide range of vision tasks when finetuned on domain-specific data. Additionally, some of these models exhibit notable prompt-based open-form (also known as zero-shot) capabilities, such as classification from text prompts (Radford et al., 2021) and segmentation from geometric prompts (e.g., points, bounding boxes, and masks) (Kirillov et al., 2023). Depending on their pre-training objectives, VFMs can act as feature extractors suitable for diverse downstream tasks. For instance, models that employ contrastive losses during training (Chen et al., 2020; Radford et al., 2021; Oquab et al., 2023), utilize low-frequency signals, and generate features that can linearly separate samples based on their semantic content (Park et al., 2022). Conversely, the pre-training objectives for MAE and SAM involve denoising masked images and instance mask segmentation, respectively. These objectives lead to the acquisition of features utilizing high-frequency signals with localization knowledge but limited semantic understanding (see Figure 4).

Maintaining and deploying separate vision models for different downstream tasks is inefficient (high memory footprint and runtime, especially on edge devices) and lacks opportunity for cross-model learning (Sanh et al., 2021). *Multitask learning* (Zhang & Yang, 2021) is a paradigm capable of addressing this issue. However, it often requires costly training and simultaneous access to all tasks (Fifty et al., 2021). Training foundation models often relies on an unsupervised or semi-

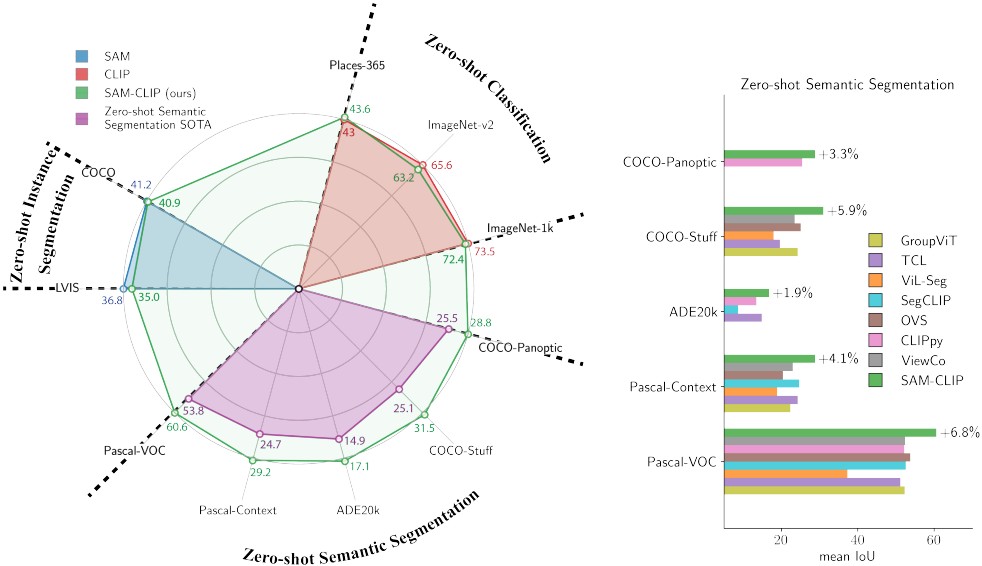

Figure 1: `SAM-CLIP` inherits most zero-shot capabilities of SAM (instance segmentation) and CLIP (classification) using a single shared backbone (**left**). Further, `SAM-CLIP` is capable of a new task, zero-shot semantic segmentation, and obtains state-of-the-art results on several benchmarks, with a large margin compared to previous models specifically designed for this task (**right**). Detailed results are provided in Tables 1 and 2.

supervised approach, requiring substantial computational resources. For example, state-of-the-art CLIP models are trained on extensive datasets, such as LAION (Schuhmann et al., 2022) and Data-Comp (Gadre et al., 2023), consuming a massive amount of computational power. Similarly, SAM's pre-training on 1.1 billion masks is computationally demanding. A multi-objective pre-training method requires comparable or more data and compute power as single objective VFM training. Additionally, there are still challenges to be addressed, such as how to best mix datasets, how to handle interfering gradients and instabilities in multi-task training (Du et al., 2019), and how to access VFM pre-training datasets that are often proprietary (Radford et al., 2021), which limit the scalability and feasibility of this approach.

To overcome these challenges, model merging has emerged as a rapidly growing area of research (Sung et al., 2023; Yadav et al., 2023). The majority of merging techniques focus on combining multiple task-specific models into a single model without requiring additional training. For instance, this can be achieved through techniques such as model weights interpolation (Ilharco et al., 2022b), parameter importance analysis (Matena & Raffel, 2022), or leveraging invariances in the models (Ainsworth et al., 2022). These techniques, on the other side, put too much stress on not using data or not performing additional training/finetuning resulting in decreased performance or lack of generalization to diverse sets of tasks (Sung et al., 2023). In this work, our goal is to merge VFMs that are trained with fundamentally different objectives, have distinct capabilities, and possibly interact with other modalities. In this setup, naive merging approaches such as weight interpolation result in significant forgetting (McCloskey & Cohen, 1989) as we show in Appendix C.

We aim to fill the gap between training-free model merging and multitask training by drawing techniques from continual learning (Li & Hoiem, 2017; Parisi et al., 2019) and knowledge distillation (Hinton et al., 2015). We treat model merging as a continual learning problem, where, given a pretrained VFM, the knowledge of a second VFM is merged without forgetting of the initial knowledge. On one side, in contrast to weight averaging techniques, we allow access to a *small part of* pretraining data or its surrogates to be replayed during the merging process. We leverage multi-task distillation on the replay data to avoid forgetting the original knowledge of pretrained VFMs during the merging process. On the other side, our merging process is significantly more efficient than traditional multitask training by requiring less than 10% of the data and computational cost compared to their original pretraining (Section 3).

We instantiate our proposed merging approach by combining SAM and CLIP into a single multi-task model, called `SAM-CLIP`, suitable for edge device deployment. This merged model inherits prompt-based zero-shot capabilities from both CLIP and SAM with minimal forgetting: specifically, zero-shot classification and image-text retrieval from CLIP, and zero-shot instance segmentation

from SAM (see Figure 1 left). Further, we illustrate that `SAM-CLIP` learns richer visual representations compared to SAM and CLIP, endowed with both spatial and semantic features, resulting in improved head-probing performance on new tasks (see Figure 4). Finally, `SAM-CLIP` shows an emerging capability of zero-shot transfer to a new task: *zero-shot semantic segmentation* thanks to combined skills inherited from SAM and CLIP. This task involves generating a segmentation mask based on a free-form text prompt. It requires both semantic understanding from text and segmentation capabilities, which are skills that `SAM-CLIP` learns from CLIP and SAM, respectively. We demonstrate that `SAM-CLIP` achieves state-of-the-art performance on zero-shot semantic segmentation in a single-stage inference setup over multiple datasets (Figure 1 right). With a compromise of a negligible drop compared to the performance of individual models on the original tasks (zero-shot classification and instance segmentation), we get a *single model* that not only masters both tasks, but also is capable of accomplishing a new task.

## 2 BACKGROUND

**Vision-Language Models** (VLMs) such as CLIP and ALIGN (Jia et al., 2021) are trained on Billion-scale, often noisy, image-text datasets. These models consist of modality-specific (image and text) encoders that produce an embedding for each modality. For a randomly sampled batch of image-text pairs, these models are trained with a contrastive objective to maximize alignment between embeddings of positive pairs of image and text. A direct application of such models is zero-shot image-text retrieval, or zero-shot classification via text prompts (Radford et al., 2021). Other works such as ViLT (Kim et al., 2021), VLMo (Bao et al., 2022), and BLIP (Li et al., 2022a) explored shared or mixed architectures between image and text modalities and enabled additional zero-shot capabilities such as Visual Question Answering (VQA) and captioning. Approaches such as LiT (Zhai et al., 2022), APE (Rosenfeld et al., 2022), and BLIP-2 (Li et al., 2023b) reduce the training cost of CLIP-like models by deploying pre-trained single-modal models. This is similar to our approach in terms of harvesting knowledge of available pre-trained models. However, we focus on *merging* vision backbones into a unified model in a multi-modal multi-encoder setup. Further, on top of representation learning abilities, we transfer zero-shot capabilities of the pre-trained models.

**Segment Anything Model** (SAM) (Kirillov et al., 2023) introduces a large-scale dataset, a model, and a training recipe to enable segmentation given a prompt. The dataset consists of triplets of an image, a geometric prompt, and a segmentation mask. SAM consists of an image encoder, a prompt encoder, and a mask decoder. SAM's image encoder is a ViT-Det (Li et al., 2022b) pretrained with MAE (He et al., 2022) objective, which is endowed with rich high-frequency localization knowledge (Park et al., 2022). The prompt-encoder gets a geometric input in the form of points, mask regions, or bounding boxes. The mask decoder gets the output of both encoders and produces a high-resolution segmentation mask. SAM is trained using a linear combination of Focal (Lin et al., 2017) and Dice (Milletari et al., 2016) losses and is capable of generating segmentation masks even when the input prompt is ambiguous/low-quality. It is noteworthy that Kirillov et al. (2023) briefly discusses a possible multi-task pre-training strategy to enable free-form text-to-mask capability, but has not released the model.

There are a few follow-up works to SAM that we briefly discuss here. HQ-SAM (Ke et al., 2023) adds an additional token and a lightweight learnable layer to a frozen SAM model to enable high-quality segmentation using a small high-quality annotated segmentation dataset. FastSAM (Zhao et al., 2023) and MobileSAM (Zhang et al., 2023) employ CNN architecture and knowledge distillation, respectively, to train smaller and faster variants of the SAM model. Unlike our work, all these methods target the same task as the original SAM and could potentially be used as the base VFM in our proposed method. Semantic-SAM (Li et al., 2023a) and SEEM (Zou et al., 2023) use semantic segmentation annotations for training to enable semantic-aware and multi-granular segmentation, hence they are not *zero-shot* semantic segmentation models. These works differ from our approach, which does not use any semantic segmentation annotations and instead gains semantic knowledge from distillation with CLIP.

**Knowledge Distillation** (KD) (Hinton et al., 2015; Buciluǎ et al., 2006) was originally proposed to train a compressed classifier (student) using knowledge accumulated in a pretrained large model (teacher). Related to our work, recent works explored distillation methods for VLMs such as EVA (Fang et al., 2023b;a), DIME-FM (Sun et al., 2023b), CLIPPING (Pei et al., 2023), and CLIP-

KD (Yang et al., 2023). They show the transfer of the same zero-shot capability of the teacher model to the student. Here, in a multi-task setup, we perform distillation and self-distillation (Furlanello et al., 2018), and demonstrate the transfer of different zero-shot capabilities (from two teachers) into a single model, as well as the emergence of new zero-shot capability specific to the student model.

**Continual Learning** (CL) Our setup is also related to Continual Learning (Parisi et al., 2019), where new knowledge is added to an existing model. The main challenge in continual learning is *catastrophic forgetting* (McClelland et al., 1995; McCloskey & Cohen, 1989) referring to the loss of previously learned knowledge due to learning new tasks. Continual Learning algorithms usually alleviate forgetting via regularization (Kirkpatrick et al., 2017; Zenke et al., 2017), experience replay (Rebuffi et al., 2017; Hayes et al., 2019), regularized replay (Chaudhry et al., 2018; Farajtabar et al., 2020), dynamic expansion (Yoon et al., 2017; Schwarz et al., 2018), and optimization based methods (Pan et al., 2020; Mirzadeh et al., 2020), among them, replay based methods proved to be simple yet very successful ones (Lomonaco et al., 2022; Balaji et al., 2020). In this work, we propose a simple recipe based on memory replay and distillation to merge VFMs with minimal forgetting.

**Zero-shot Semantic Segmentation** task aims to predict a dense segmentation mask given a text prompt in an open form, without prior knowledge of specific object classes of interest or any finetuning. Recent approaches to open-vocabulary segmentation deploy image-text pairs datasets and pretrained VLMs such as CLIP and their internal representations to obtain dense segmentation masks, for example GroupViT (Xu et al., 2022), ViewCo (Ren et al., 2023), CLIPpy (Ranasinghe et al., 2023), ViL-Seg (Liu et al., 2022), OVS (Xu et al., 2023), TCL (Cha et al., 2023), and SegCLIP (Luo et al., 2023). In this work, we do not directly use any text data. Instead, all text semantic knowledge is derived from a pretrained CLIP. An alternative approach is to deploy existing models, without any training, and generate segmentation masks using multiple backbones in a multi-stage setup. For example, one can run SAM to get several object proposals and run each through CLIP for semantic classification (Liu et al., 2023). Some recent works (Karazija et al., 2023; Wang et al., 2023) use internal attention maps of conditional vision generative models such as StableDiffusion (Rombach et al., 2022) to obtain segmentation masks. While these approaches are training-free, they require several stages with complex processing, multiple vision encoders, and many forward passes, making their deployment for edge devices limited.

**Merging Models** techniques aim to combine the capability of different models by simple interpolation operations such as weight averaging (Wortsman et al., 2022) and task arithmetic (Ilharco et al., 2022b). Recently there's abundance of such techniques (Choshen et al., 2022; Matena & Raffel, 2022; Muqeeth et al., 2023; Wu et al., 2023; Ilharco et al., 2022a; Stoica et al., 2023; Khanuja et al., 2021; Bai et al., 2022) employing different weight schemes and parameter sensitivity and importance. The way we train `SAM-CLIP`, can be regarded as a data-dependent merging approach where the knowledge of the models is combined by repeatedly reminding them of their original behavior via replay, while the optimization algorithm explores the parameter space to find an optimum.

## 3 PROPOSED APPROACH

In this section, we explain our approach for efficiently merging pretrained VFMs. We start with a base VFM, then transfer knowledge from other auxiliary VFMs to it with minimal forgetting. We assume that each VFM possesses a vision encoder, and potentially other modality encoders, as well as task-specific decoders/heads. Our goal is to combine the vision encoders into a single backbone such that it can be used in conjunction with other modality encoders, which remain frozen.

To focus our exposition, we constrain our discussion to the specific case where SAM serves as the base VFM, while a CLIP model serves as the auxiliary VFM. This pair presents an intriguing combination, as both models have been successfully deployed in diverse tasks and exhibit complementary capabilities. SAM excels in localization and high-resolution image segmentation but has limitations in semantic understanding. Conversely, CLIP offers a powerful image backbone for semantic understanding. We demonstrate it by several probing experiments (see Figure 4). Potentially, one could start with CLIP as the base VFM and merge knowledge of SAM to it. However, existing pretrained CLIP ViT models are inefficient in dealing with high-resolution images that are used for SAM training. Hence, we choose SAM as the base model and inherit its ViT-Det structure that can process high-resolution inputs efficiently.

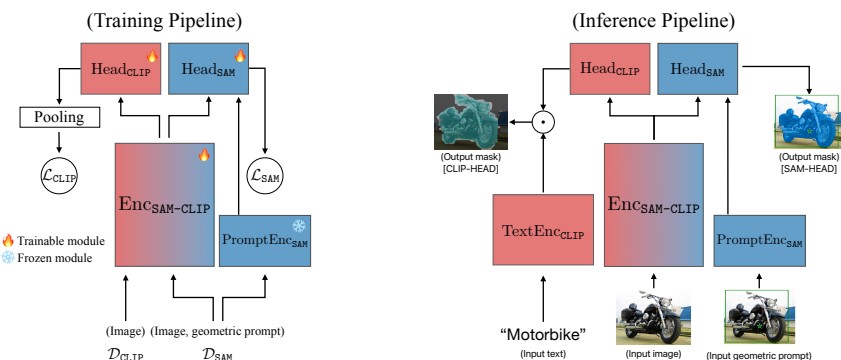

Figure 2: Multi-head architecture of `SAM-CLIP`. **Left**: the training pipeline where we perform multi-task distillation from CLIP and SAM teacher models on $\mathcal{D}_{\text{CLIP}}$ and $\mathcal{D}_{\text{SAM}}$ datasets, respectively. **Right**: shows our inference pipeline where with a single backbone we can perform multiple promptable tasks: classification, instance segmentation, and semantic segmentation. $\odot$ denotes the inner product between text embedding and image patch embeddings.

We assume access to limited subsets of datasets (or their proxies) used to train the base and auxiliary VFMs, which function as memory replay in our CL setup. These are denoted as $\mathcal{D}_{\text{SAM}}$ and $\mathcal{D}_{\text{CLIP}}$, respectively with details provided in Section 4.1.

We employ a multi-head architecture, illustrated in Figure 2. Our base VFM, SAM, has an image encoder ($\text{Enc}_{\text{SAM}}$), a prompt encoder ($\text{PromptEnc}_{\text{SAM}}$), and a light mask decoder ($\text{MaskDec}_{\text{SAM}}$). The auxiliary VFM, CLIP, has an image encoder ($\text{Enc}_{\text{CLIP}}$) and a text encoder ($\text{TextEnc}_{\text{CLIP}}$). Our goal is to merge both image encoders to a single backbone called $\text{Enc}_{\text{SAM-CLIP}}$ which is initialized by $\text{Enc}_{\text{SAM}}$. Further, we consider lightweight heads corresponding to each VFM, namely, $\text{Head}_{\text{SAM}}$ and $\text{Head}_{\text{CLIP}}$. $\text{Head}_{\text{SAM}}$ is initialized with $\text{MaskDec}_{\text{SAM}}$ and $\text{Head}_{\text{CLIP}}$ is initialized with random weights (since CLIP does not come with a head that we can deploy). We deploy other modality encoders (i.e., $\text{PromptEnc}_{\text{SAM}}$ and $\text{TextEnc}_{\text{CLIP}}$) with no change (frozen).

As a baseline merging approach, we perform KD on $\mathcal{D}_{\text{CLIP}}$ utilizing a cosine distillation loss (Grill et al., 2020):

$$\mathcal{L}_{\text{CLIP}} = \mathbb{E}_{\boldsymbol{x} \sim \mathcal{D}_{\text{CLIP}}} \left[ 1 - \phi^{\text{Pooling}}(\text{Head}_{\text{CLIP}}(\text{Enc}_{\text{SAM-CLIP}}(\boldsymbol{x})))^T \text{Enc}_{\text{CLIP}}(\boldsymbol{x}) \right], \quad (1)$$

where $\phi^{\text{Pooling}}$ is a spatial pooling operator that gets patch-level features from $\text{Head}_{\text{CLIP}}$ and produces a normalized image-level embedding. In this setup, parameters of both $\text{Head}_{\text{CLIP}}$ and $\text{Enc}_{\text{SAM-CLIP}}$ are learnable, while the CLIP encoder, $\text{Enc}_{\text{CLIP}}$, is frozen and used as a teacher. While this infuses SAM with CLIP's semantic abilities, it incurs at the cost of catastrophic forgetting of SAM's original capabilities. Further, we show that training-free mitigative methods against catastrophic forgetting, such as Wise-FT (Wortsman et al., 2022), to be ineffective in our context of VFM merging, as demonstrated in section C.

To address these challenges, we propose a rehearsal-based multi-task distillation. This serves two primary goals: 1) facilitate the efficient transfer of knowledge from the auxiliary VFM to the base model, and 2) preserve the original capabilities of the base model. Inspired by Kumar et al. (2022), we consider a two-stage training: head-probing and multi-task distillation. An optional stage of resolution adaptation can be appended if the multiple heads are trained under different resolutions, which is the case in our experiment of merging SAM and CLIP. See Section 4.1 for details about resolution adaptation.

**I. Head probing:** In this stage, we first freeze the image backbone, $\text{Enc}_{\text{SAM-CLIP}}$, and only train $\text{Head}_{\text{CLIP}}$ with the loss in Equation (1). Intuitively, with this approach, we first learn some reasonable values for parameters of $\text{Head}_{\text{CLIP}}$ (which is initialized randomly) before allowing any change in $\text{Enc}_{\text{SAM-CLIP}}$ that is prone to forgetting.

**II. Multi-task distillation:** In this stage, we allow all heads as well as our image encoder to be learnable. We perform a multi-task training on $\mathcal{L}_{\text{CLIP}} + \lambda \mathcal{L}_{\text{SAM}}$, with:

$$\mathcal{L}_{\text{SAM}} = \mathbb{E}_{(\boldsymbol{x},\boldsymbol{g}) \sim \mathcal{D}_{\text{SAM}}} \mathcal{L}_{\text{FD}}(\text{Head}_{\text{SAM}}(\text{Enc}_{\text{SAM-CLIP}}(\boldsymbol{x}), \text{PromptEnc}_{\text{SAM}}(\boldsymbol{g})), \boldsymbol{z}), \quad (2)$$

where, $\boldsymbol{x}$ is a raw image, $\boldsymbol{g}$ is a geometric prompt, $\boldsymbol{z} = \text{MaskDec}_{\text{SAM}}(\text{Enc}_{\text{SAM}}(\boldsymbol{x}))$ is segmentation mask score produced by frozen SAM teacher, and $\mathcal{L}_{\text{FD}}$ refers to a linear combination of Focal (Lin

Table 1: Zero-shot evaluations on classification and instance segmentation tasks, comparing `SAM-CLIP` with state-of-the-art models that use the ViT-B architecture. `SAM-CLIP` demonstrates minimal forgetting compared to the baseline FMs on their original tasks.

| Model | Training Data | 0-Shot Classification (%) | | | 0-Shot Instance Seg. (mAP) | |
|---|---|---|---|---|---|---|
| | | ImageNet | ImageNet-v2 | Places-365 | COCO | LVIS |
| SAM (Kirillov et al., 2023) | SA-1B | - | - | - | 41.2 | 36.8 |
| CLIP (Radford et al., 2021) | OpenAI-400M | 68.3 | 62.6 | 42.2 | - | - |
| CLIP (Cherti et al., 2023) | LAION-2B | 71.1 | 61.7 | 43.4 | - | - |
| CLIP (Gadre et al., 2023) | DataComp-1B | 73.5 | 65.6 | 43.0 | - | - |
| SAM-CLIP (Ours) | Merged-41M | 72.4 | 63.2 | 43.6 | 40.9 | 35.0 |

et al., 2017) and Dice (Milletari et al., 2016) used in the original SAM training adapted for distillation. We train on $\mathcal{D}_{\text{SAM}} \cup \mathcal{D}_{\text{CLIP}}$ with total loss of $\mathcal{L}_{\text{CLIP}} + \lambda \mathcal{L}_{\text{SAM}}$. During training, each batch has some samples from $\mathcal{D}_{\text{CLIP}}$ and some form $\mathcal{D}_{\text{SAM}}$, which contribute to $\mathcal{L}_{\text{CLIP}}$ and $\mathcal{L}_{\text{SAM}}$, respectively (i.e., samples from CLIP dataset do not contribute to SAM loss and vice versa). To encourage less forgetting, we use an order of magnitude smaller learning rate for parameters of $\text{Enc}_{\text{SAM-CLIP}}$ and $\text{Head}_{\text{SAM}}$ compared to $\text{Head}_{\text{CLIP}}$ at this stage.

## 4 EXPERIMENTS

### 4.1 IMPLEMENTATION DETAILS

Our design choices, as explained below, aim to balance the trade-off between learning from CLIP (zero-shot classification) and retaining SAM's knowledge (instance segmentation).

**Model Architecture.** We employ the ViT-B/16 version of the Segment Anything Model (SAM) as our base architecture (Kirillov et al., 2023), comprising 12 transformer layers. To integrate CLIP capabilities, we append a lightweight CLIP head consisting of 3 transformer layers to the SAM backbone. The patch token outputs from this CLIP head undergo a pooling layer to produce an image-level embedding, akin to the role of the CLS token output in ViT models. We adopt max-pooling since we observe that it can lead to better zero-shot classification and semantic segmentation performance of `SAM-CLIP` than average pooling. It is noteworthy that max-pooling has been found to be able to encourage the learning of spatial visual features (Ranasinghe et al., 2023). With the pooling layer, the CLIP head can output an embedding for the whole image, which can be aligned with a text embedding just like the original CLIP model (Radford et al., 2021).

**Dataset Preparation.** For the CLIP distillation, we merge images from several datasets: CC3M (Sharma et al., 2018), CC12M (Changpinyo et al., 2021), YFCC-15M (Radford et al., 2021) (a curated subset of YFCC-100M (Thomee et al., 2016) by OpenAI) and ImageNet-21k (Ridnik et al., 2021). This forms our $\mathcal{D}_{\text{CLIP}}$ containing 40.6M unlabeled images. For the SAM self-distillation, we sample 5.7% subset from the SA-1B dataset to form $\mathcal{D}_{\text{SAM}}$, which originally comprises 11M images and 1.1B masks. We randomly select 1% of $\mathcal{D}_{\text{CLIP}}$ and $\mathcal{D}_{\text{SAM}}$ as validation sets. Overall, we have 40.8M images for training, which we term as Merged-41M in this work.

**Training.** As we discussed in Sec. 3, the training is conducted in two phases to optimize convergence, in a "*probing then full finetuning*" style. The first stage of CLIP-head probing takes 20 epochs on $\mathcal{D}_{\text{CLIP}}$, while the backbone is kept frozen. Here, the teacher model is the OpenCLIP (Ilharco et al., 2021) ViT-L/14 trained on the DataComp-1B dataset (Gadre et al., 2023). In the second stage (16 epochs), we unfreeze the backbone $\text{Enc}_{\text{SAM-CLIP}}$ and proceed with joint fine-tuning together with $\text{Head}_{\text{CLIP}}$ and $\text{Head}_{\text{SAM}}$, incorporating both CLIP and SAM distillation losses at the ratio of 1:10. The original SAM ViT-B model serves as the teacher in SAM loss. Further, the learning rates applied to $\text{Enc}_{\text{SAM-CLIP}}$ and $\text{Head}_{\text{SAM}}$ are 10 times smaller than that of $\text{Head}_{\text{CLIP}}$ in order to reduce the forgetting of the original SAM abilities. Besides, we adopt a mixed input resolution strategy for training. A notable difference between SAM and CLIP is their pre-training resolution. SAM is trained and works best on 1024px resolution while often lower resolutions (e.g., 224/336/448px) are adopted for CLIP training and inference (Radford et al., 2021; Cherti et al., 2023; Sun et al., 2023a). Hence, we employ variable resolutions of 224/448px for the CLIP distillation via the variable batch

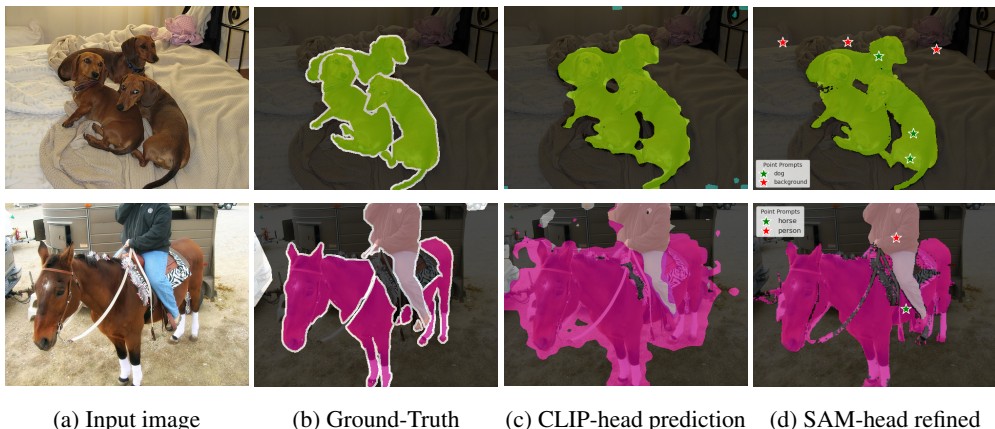

| (a) Input image | (b) Ground-Truth | (c) CLIP-head prediction | (d) SAM-head refined |

Figure 3: Demo on zero-shot semantic segmentation. Passing an input image through the image encoder, $\text{Head}_{\texttt{CLIP}}$ can predict a semantic segmentation mask, and $\text{Head}_{\texttt{SAM}}$ can refine it to a more fine-grained mask with auto-generated geometric prompts.

sampler approach of Mehta et al. (2022), while SAM distillation utilizes a 1024px resolution in accordance with SAM's original training guidelines (Kirillov et al., 2023). In every optimization step, we form a batch of 2048 images from $\mathcal{D}_{\texttt{CLIP}}$ and 32 images (each with 32 mask annotations) from $\mathcal{D}_{\texttt{SAM}}$ and perform training in a multi-task fashion (see Appendix A for more details).

**Resolution Adaption.** After the two training stages, $\texttt{SAM-CLIP}$ can accomplish CLIP tasks (e.g., zero-shot classification) using the CLIP-head under 224/336/448px, and run inference with the SAM-head under 1024px. However, if one wants to apply the two heads together on a single input image for certain tasks (we present a demo of this in Sec. 4.4), it would be inefficient to pass the image twice to the image encoder with two resolutions for the two heads respectively. To remedy this issue, we adapt the CLIP head for 1024px input using a very short and efficient stage of finetuning: freezing the image encoder and only finetuning the CLIP-head with $\mathcal{L}_{\texttt{CLIP}}$ for 3 epochs (it is the same as the first stage of training, which is also CLIP-head probing) under variable resolutions of 224/448/1024px. *Note:* resolution upscaling strategies are prevalent in CLIP training: Radford et al. (2021); Sun et al. (2023a); Li et al. (2023c) show it is more efficient than training with high resolution from the beginning.

**More Details** about implementation and training are presented in the Appendix A.

## 4.2   ZERO-SHOT EVALUATIONS

**CLIP Task:  Zero-Shot Image Classification.** To examine the CLIP-related capabilities of $\texttt{SAM-CLIP}$, we evaluate it with zero-shot image classification on ImageNet (Deng et al., 2009), ImageNet-v2 (Recht et al., 2019) and Places365 (Zhou et al., 2017), under image resolution of 336px. We use the text templates as Radford et al. (2021) utilizing the textual embeddings from the text encoder of $\texttt{SAM-CLIP}$ (which is kept frozen from our CLIP teacher) to perform zero-shot classification without any finetuning. The evaluation results are presented in Table 1. Employing a ViT-B architecture, our model achieves zero-shot accuracy comparable to the state-of-the-art CLIP ViT-B models pretrained on LAION-2B (Schuhmann et al., 2022) and DataComp-1B (Gadre et al., 2023) (both released by Ilharco et al. (2021)), over the three datasets. These results validate the efficacy of our merging approach in inheriting CLIP's capabilities. *Note:* We observe that $\texttt{SAM-CLIP}$ benefits from a 336px resolution for zero-shot image classification, whereas the baseline CLIP models do not, as they were trained at a 224px resolution (the reported results of baseline CLIP models in Table 1 are evaluated at 224px). The evaluation results of $\texttt{SAM-CLIP}$ at 224px vs. 336px resolutions are provided in Appendix A.

**SAM Task: Zero-Shot Instance Segmentation.** For the SAM component of $\texttt{SAM-CLIP}$, we evaluate its performance in instance segmentation, a task at which the original SAM model excels (Kirillov et al., 2023), with COCO (Lin et al., 2014) and LVIS (Gupta et al., 2019) datasets. Following the original practices of Kirillov et al. (2023), we first generate object detection bounding boxes using a ViT-Det model (ViT-B version) (Li et al., 2022b). These bounding boxes act as geometric

Table 2: Zero-shot semantic segmentation performance comparison with recent works. **Note:** The results of SAM-CLIP below are obtained by using the CLIP-head only. The results with SAM-head refinement are provided in Table 5. ([†]SegCLIP is trained on COCO data, so it is not zero-shot transferred to COCO-Stuff.)

| Model | Arch | Training Data | 0-Shot Semantic Segmentation (mIoU %) | | | | |
|---|---|---|---|---|---|---|---|
| | | | Pascal VOC | Pascal-Context | ADE20k | COCO-Stuff | COCO-Panoptic |
| GroupViT (Xu et al., 2022) | ViT-S | Merged-26M | 52.3 | 22.4 | - | 24.3 | - |
| ViewCo (Ren et al., 2023) | ViT-S | Merged-26M | 52.4 | 23.0 | - | 23.5 | - |
| ViL-Seg (Liu et al., 2022) | ViT-B | CC12M | 37.3 | 18.9 | - | 18.0 | - |
| OVS (Xu et al., 2023) | ViT-B | CC4M | 53.8 | 20.4 | - | 25.1 | - |
| CLIPpy (Ranasinghe et al., 2023) | ViT-B | HQITP-134M | 52.2 | - | 13.5 | - | 25.5 |
| TCL (Cha et al., 2023) | ViT-B | CC3M+CC12M | 51.2 | 24.3 | 14.9 | 19.6 | - |
| SegCLIP (Luo et al., 2023) | ViT-B | CC3M+COCO | 52.6 | 24.7 | 8.7 | 26.5[†] | - |
| SAM-CLIP (CLIP-head) | ViT-B | Merged-41M | **60.6** | **29.2** | **17.1** | **31.5** | **28.8** |

Table 3: Head probing evaluations on semantic segmentation datasets, comparing our model with SAM and CLIP that use the ViT-B architecture. Avg is the average evaluation results of three heads.

| | Training Data | Pascal VOC | | | | ADE20k | | | |
|---|---|---|---|---|---|---|---|---|---|
| Model | | Linear | DeepLabv3 | PSPNet | Avg | Linear | DeepLabv3 | PSPNet | Avg |
| SAM | SA-1B | 46.6 | 69.9 | 71.2 | 62.6 | 26.6 | 32.8 | 36.2 | 31.9 |
| CLIP | DataComp-1B | 70.7 | 78.9 | 79.7 | 76.4 | 36.4 | 39.4 | 40.7 | 38.8 |
| SAM-CLIP | Merged-41M | **75.0** | **80.3** | **81.3** | **78.8** | **38.4** | **41.1** | **41.7** | **40.4** |

prompts for SAM's prompt encoder, which then predicts masks for each object instance. The evaluation results of SAM-CLIP and the original SAM ViT-B are provided in Table 1 (both under 1024px resolution), showing that SAM-CLIP is very close to SAM on the two benchmarks, not suffering from catastrophic forgetting during training.

**Zero-Shot Transfer to Semantic Segmentation.** We extend our evaluation to (text-prompted) zero-shot semantic segmentation over 5 datasets, Pascal VOC (Everingham et al., 2010), Pascacl Context (Mottaghi et al., 2014), ADE20k (Zhou et al., 2019), COCO-Stuff (Caesar et al., 2018) and COCO-Panoptic (Kirillov et al., 2019; Lin et al., 2014). We adopt a common evaluation protocol for this task: i) each input image is resized to $448 \times 448$px and pass to the image encoder and CLIP-head of SAM-CLIP to obtain $28 \times 28$ patch features; ii) OpenAI's 80 pre-defined CLIP text templates are employed to generate textual embeddings for each semantic class, and these embeddings act as mask prediction classifiers and operate on the patch features from the CLIP head; iii) we linearly upscale the mask prediction logits to match the dimensions of the input image. Evaluation results of SAM-CLIP and previous zero-shot models over the five datasets are demonstrated in Fig. 2. Notably, SAM-CLIP establishes new state-of-the-art performance on all 5 datasets, with a significant margin over past works. More details are provided in Appendix B.

## 4.3 HEAD-PROBING EVALUATIONS ON LEARNED REPRESENTATIONS

By merging the SAM and CLIP models, we anticipate that the resultant model will inherit advantages at the representation level from both parent models. Specifically, SAM excels at capturing low-level spatial visual details pertinent to segmentation tasks, while CLIP specializes in high-level semantic visual information encompassing the entire image. We hypothesize that the merged model combines these strengths, thereby enhancing its utility in broad range of downstream vision tasks. To investigate this hypothesis, we conduct head-probing (i.e., learn a task specific head with a frozen image backbone) evaluations on SAM, CLIP, and SAM-CLIP, utilizing different segmentation head structures (linear head, DeepLab-v3 (Chen et al., 2017) and PSPNet (Zhao et al., 2017)) across two semantic segmentation datasets, Pascal VOC and ADE20k. The results are presented in Table 3. We observe that SAM representations do not perform as well as those of CLIP for tasks that require semantic understanding, even for semantic segmentation task. However, SAM-CLIP outperforms both SAM and CLIP across different head structures and datasets, thereby confirming its superior visual feature representation capabilities.

Besides, we apply linear probing to these models for image classification tasks on two datasets, ImageNet and Places365. Results in Table 4 show that SAM-CLIP attains comparable performance with CLIP, implying that the image-level representation of SAM-CLIP is also well-learned. All head probing evaluation results are visualized in Figure 4 to deliver messages more intuitively.

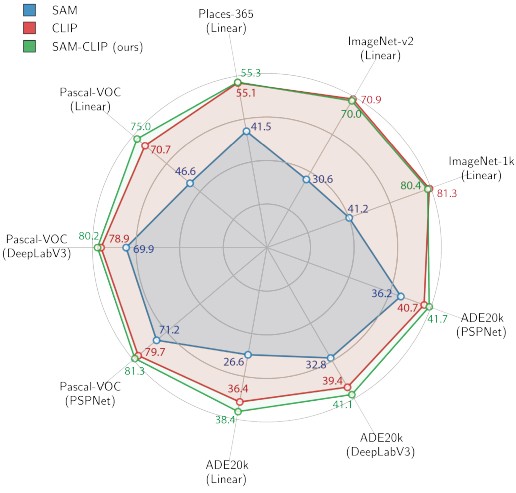

Figure 4: Representation learning comparison. Head-probing evaluation of each vision backbone for classification and semantic segmentation tasks. SAM-CLIP learns richer visual features compared to SAM and CLIP.

Table 4: Linear probing evaluations on image classification datasets with ViT-B models.

| Model | Linear Probing | |
| | ImageNet | Places365 |
| --- | --- | --- |
| SAM | 41.2 | 41.5 |
| CLIP (DataComp1B) | **81.3** | 55.1 |
| CLIP (LAION-2B) | 79.6 | 55.2 |
| SAM-CLIP | 80.5 | **55.3** |

Table 5: Composing both CLIP and SAM heads of SAM-CLIP for zero-shot semantic segmentation on Pascal VOC.

| Method | Resolution | mIoU |
| --- | --- | --- |
| CLIP head only | 448px | 60.6 |
| CLIP+SAM heads | 1024px | **66.0** |

## 4.4 COMPOSING BOTH CLIP AND SAM HEADS FOR BETTER SEGMENTATION

Given that SAM-CLIP is a multi-task model with SAM and CLIP heads, one would naturally ask if the two heads can work together towards better performance on some tasks. Here, we showcase that a simple composition of the CLIP and SAM heads can lead to better zero-shot semantic segmentation. Specifically, we resize the input image to 1024px and pass it through $\text{Enc}_{\text{SAM-CLIP}}$, and use the CLIP head to generate low-resolution mask prediction ($32 \times 32$) using text prompts. Then, we generate some point prompts from the mask prediction (importance sampling based on the mask prediction confidence), and pass the mask prediction and point prompts together to the prompt encoder module as geometric prompts. Finally, $\text{Head}_{\text{SAM}}$ takes embeddings from both the prompt encoder and the image encoder to generate high-resolution mask predictions ($256 \times 256$) as shown in Figure 2 (right). Examples of this pipline are shown in Figure 3. One can clearly observe that the refined segmentation by the SAM-head is more fine-grained. The implementation details about this pipeline is discussed in Appendix B.

Note that this pipeline requires *only one forward pass* on $\text{Enc}_{\text{SAM-CLIP}}$ with 1024px resolution. For fair comparison, in Table 1 and Figure 1 we report SAM-CLIP zero-shot segmentation performance with 448px resolution using $\text{Head}_{\text{CLIP}}$ only. Using our high-resolution pipeline we obtain further gain in zero-shot semantic segmentation as shown in Table 5.

## 5 CONCLUSION

We discussed merging publicly available vision foundation models, as digested sources of visual knowledge, into a single unified architecture. We proposed a simple and efficient recipe based on multi-task distillation and memory rehearsal. Specifically, we instantiated our proposed approach to merge SAM and CLIP vision foundation models, and introduced SAM-CLIP. SAM and CLIP have complementary vision capabilities: one is good on spatial understanding, while the other excels on semantic understanding of images. We demonstrate multiple benefits as a result of our proposed approach: 1) We obtain a single vision backbone with minimal forgetting of zero-shot capabilities of the original models, suitable for edge device deployment. 2) We demonstrate the merged model produces richer representations utilizable for more diverse downstream tasks when compared to original models in a head-probing evaluation setup. 3) The merged model demonstrates synergistic new zero-shot capability thanks to complementary inherited skills from the parent models. Specifically, we show that SAM-CLIP obtains state-of-the-art performance on zero-shot semantic segmentation by combining semantic understanding of CLIP and localization knowledge of SAM.

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

## A    MORE EXPERIMENTAL DETAILS

**Software**    We built our codebase using PyTorch (Paszke et al., 2019) and the CVNets framework (Mehta et al., 2022). The evaluation code for instance segmentation relies on the publicly released codebases from Kirillov et al. (2023) and Li et al. (2022b).

**Hardware**    We conducted all experiments on servers equipped with $8\times$A100 GPUs. For training our models, we most employed multi-node training across four $8\times$A100 servers. The local batch size per server is one-fourth of the global batch size.

**CLIP Head Structure**    We initialized each transformer layer of the CLIP head using parameters from the last transformer layer of SAM ViT-B, as we found this approach to expedite training compared to random initialization. Following the implementation of CLIP-ConvNeXt in Ilharco et al. (2021) (the only OpenCLIP model that uses a pooling layer instead of a CLS token), we incorporated a LayerNorm layer subsequent to the pooling layer. After applying LayerNorm, we use a shallow MLP with two hidden layers to project the features into the text-embedding space, consistent with the approach in Rosenfeld et al. (2022).

**Hyperparameters**    We employ AdamW optimizers (Loshchilov & Hutter, 2017) with a learning rate of $8 \times 10^{-4}$ (consistent with SAM training (Kirillov et al., 2023)) during the first training stage (head probing) for 20 epochs. This rate is reduced to $4 \times 10^{-5}$ during the second stage (joint distillation) for 16 epochs. It should be noted that we apply a learning rate multiplier of 0.1 to the backbone and SAM head in the second stage to mitigate forgetting. The learning rate in the resolution adaptation stage (3 epochs) remains the same as in the first stage. The global image batch size for CLIP distillation is 2048, and for SAM distillation, it is 32 (i.e., 32 images from the SA-1B dataset (Kirillov et al., 2023)). In the latter case, we randomly sample 32 masks for each image.

**Multi-Task Distillation**    Our training process consists of two stages: 1) Head probing to learn parameters of $\text{Head}_{\text{CLIP}}$ that are initialized randomly, and 2) Joint training of the $\text{Head}_{\text{SAM}}$, $\text{Head}_{\text{CLIP}}$, and the ViT backbone $\text{Enc}_{\text{SAM-CLIP}}$ using a multi-task distillation loss.

In the first stage, only the $\text{Head}_{\text{CLIP}}$ is trainable, and it is trained using a single CLIP distillation loss (cosine distance between embeddings as in Equation (1)). At this stage, all image batches are sampled only from $\mathcal{D}_{\text{CLIP}}$. This stage involves training for a fixed duration of 20 epochs without early stopping. The motivation for this step is to have a warm start for the $\text{Head}_{\text{CLIP}}$ in the next stage where we also allow modifying the backbone, similar to Kumar et al. (2022).

In the second stage, the $\text{Head}_{\text{SAM}}$ and the ViT backbone $\text{Enc}_{\text{SAM-CLIP}}$ become also trainable, and we have a multi-task objective: CLIP Distillation Equation (1) and SAM self-distillation Equation (2). The balance between the losses is determined by the coefficient $\lambda$, which we picked to optimize the trade-off between learning semantic knowledge from CLIP and forgetting SAM's segmentation knowledge. We experimented with $\lambda = 1, 10, 100$, and found that $\lambda = 10$ offers the best trade-off between mitigating the forgetting of SAM's ability and learning CLIP's ability.

Each training step for the second stage is performed as follows:

- Sample a batch of 2048 images from $\mathcal{D}_{\text{CLIP}}$. 2048 is determined based on available total GPU memory. Run the forward pass, and compute gradients backward from $\mathcal{L}_{\text{CLIP}}$ (note that only parameters of the $\text{Head}_{\text{CLIP}}$ and $\text{Enc}_{\text{SAM-CLIP}}$ will get gradients after this step).

- Sample a batch of 32 images from $\mathcal{D}_{\text{SAM}}$. 32 is determined based on available total GPU memory. Run the forward pass, and compute gradients backward from $\mathcal{L}_{\text{SAM}}$ (note that only parameters of the $\text{Head}_{\text{SAM}}$ and $\text{Enc}_{\text{SAM-CLIP}}$ will get gradients after this step).

- Apply one optimization step (note that at this point, the parameters of the $\text{Enc}_{\text{SAM-CLIP}}$ have accumulated gradients from both of the above two steps).

We early-stop after 16 epochs (out of a full training length of 20 epochs) as we observed more forgetting (as measured by instance segmentation performance on the COCO dataset) after the 16th epoch.

**Loss Coefficients**   We empirically determined the loss coefficient ratio of 1:10 for the CLIP and SAM distillation losses from three options: 1:1, 1:10, and 1:100. This ratio provides the best trade-off between mitigating SAM's ability to forget and fostering the learning of CLIP's ability. Specifically, a ratio of 1:1 leads to greater forgetting of SAM's original ability (as measured by the performance drop in instance segmentation on COCO), while ratios of 1:10 and 1:100 maintain it relatively well. However, a ratio of 1:100 impedes the learning of CLIP's ability (as measured by zero-shot accuracy on ImageNet). Therefore, we ultimately selected the ratio of 1:10.

**Image Resolution for Zero-Shot Classification**   In Table 1, we report the evaluation results for both `SAM-CLIP` and CLIP models using the 224px image resolution. However, we found that `SAM-CLIP` benefits from the 336px resolution, whereas the performance of CLIP models deteriorates (they exhibit worse accuracy). The 336px results for `SAM-CLIP` are incorporated into the diagram in Figure 1. We provide a comparison between the 224px and 336px resolutions for `SAM-CLIP` in Table 6.

Table 6: Different input resolutions for zero-shot image classification.

| Resolution | ImageNet | ImageNet-v2 | Places365 |
|---|---|---|---|
| **224px** | 71.7 | 63.2 | 43.4 |
| **336px** | 72.4 | 63.2 | 43.6 |

### A.1   COMPARATIVE ANALYSIS OF SEGMENTATION IN SAM VS. SAM-CLIP

**Comparison on Instance Segmentation**   Table 1 provides a quantitative comparison of SAM and `SAM-CLIP` on two instance segmentation datasets (COCO and LVIS), showing that `SAM-CLIP` maintains comparable performance to SAM. To give readers a more intuitive understanding of the segmentation quality of SAM versus `SAM-CLIP`, we present two examples in Figure 5. These examples demonstrate that, given the same geometric prompts (bounding box and point prompt), the segmentation masks predicted by SAM and `SAM-CLIP` are quite similar, with slight differences. This suggests that the segmentation quality of `SAM-CLIP` is indeed comparable to that of SAM.

**Comparison on Semantic Segmentation**   Figure 3 illustrates the semantic segmentation outputs of `SAM-CLIP`, featuring both CLIP-head segmentation predictions and SAM-head refined segmentation predictions. Specifically, the SAM-head refinement utilizes the CLIP-head output and some auto-generated point prompts from this output. The same point prompts are fed to SAM ViT-B, with its segmentation prediction shown in Figure 6. It is evident that SAM's prediction typically segments only a sub-part of the object indicated by the point prompts, instead of segmenting the entire semantic object class (e.g., "dog," "horse," "human"). This indicates that the CLIP-head of `SAM-CLIP` is essential for semantic segmentation, as it provides semantic understanding to the SAM-head of `SAM-CLIP`. In contrast, the point prompting approach used in SAM (Kirillov et al., 2023) is insufficient for semantic segmentation. Furthermore, point prompting requires human-provided points, making it not qualified for *zero-shot* semantic segmentation. In contrast, `SAM-CLIP` requires only text prompts for each object class (e.g., "dog," "horse," "human") to automatically generate semantic segmentation masks (the point prompts are auto-generated from the CLIP-head output in our pipeline).

## B   INFERENCE EXPERIMENTS

**CLIP and SAM Tasks**   The inference process for zero-shot classification is identical to that of the original CLIP (Radford et al., 2021; Cherti et al., 2023). The evaluation of zero-shot instance segmentation also exactly follows the protocol outlined in Kirillov et al. (2023). The image resolutions for classification and instance segmentation tasks are set at 224px and 1024px, respectively.

**Zero-Shot Semantic Segmentation**   For zero-shot semantic segmentation, we largely adhere to the practices outlined by Ranasinghe et al. (2023). We insert the class names into 80 prompt templates created by Radford et al. (2021) and obtain text embeddings using the text encoder. Next,

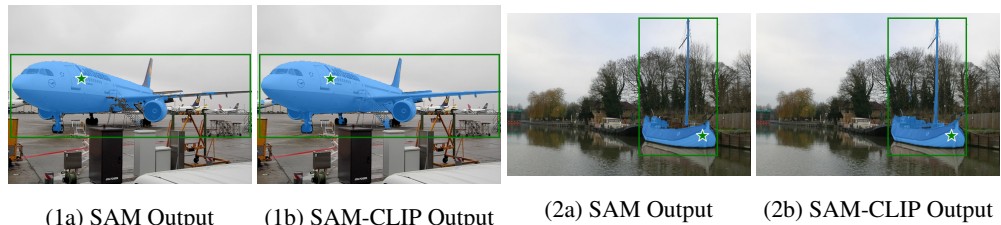

| (1a) SAM Output | (1b) SAM-CLIP Output | (2a) SAM Output | (2b) SAM-CLIP Output |

Figure 5: Comparison of instance segmentation between SAM and `SAM-CLIP` . The same images, along with geometric prompts (bounding box and point), are provided to both SAM and `SAM-CLIP` , and their respective model outputs are displayed above. While the outputs of SAM and `SAM-CLIP` exhibit slight differences, they are overall quite similar.

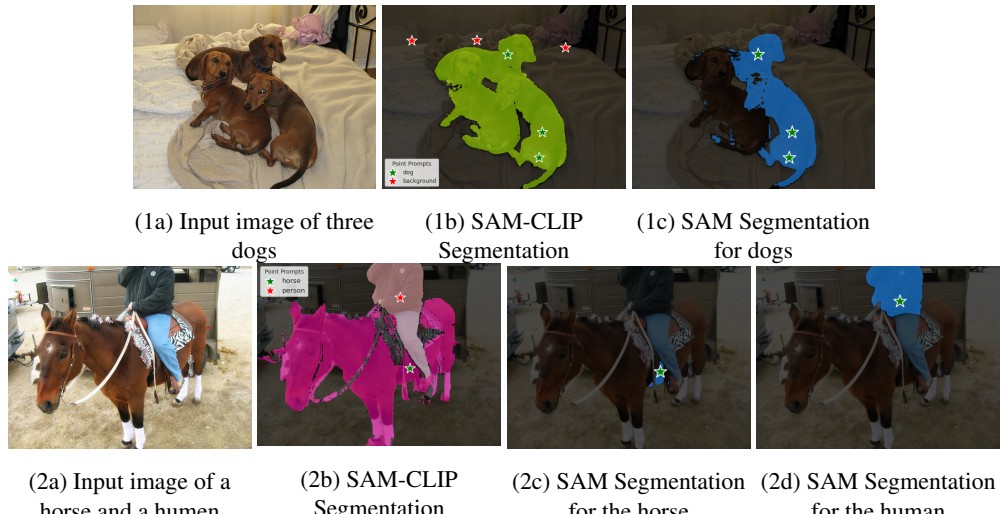

| (1a) Input image of three dogs | (1b) SAM-CLIP Segmentation | (1c) SAM Segmentation for dogs |

| (2a) Input image of a horse and a humen | (2b) SAM-CLIP Segmentation | (2c) SAM Segmentation for the horse | (2d) SAM Segmentation for the human |

Figure 6: Comparison of SAM vs. `SAM-CLIP` for semantic segmentation on two images. The segmentation of `SAM-CLIP` is obtained by: i) using CLIP-head output (i.e., coarse-grained prediction masks) to generate point prompts automatically, and ii) passing the CLIP-head output and point prompts to the SAM-head to generate final fine-grained prediction masks. For SAM, the same point prompts for each class ("dog", "human", "human") are passed to its prompt encoder to generate a segmentation mask.

we compute the cosine similarity between each text embedding and the corresponding patch feature (the output of the CLIP head). The class with the highest cosine similarity is selected as the predicted class for each patch. We then resize the patch class predictions to match the original image dimensions and calculate mIoU scores. The evaluation resolution is maintained at 448px for fair comparison with previous methods.

**Composing CLIP and SAM Heads** To combine both CLIP and SAM heads for zero-shot semantic segmentation, we first resize the image to 1024px and run the CLIP head to obtain mask predictions (i.e., logits) for each class. Subsequently, we pass the mask prediction corresponding to each class to the prompt encoder, along with 1-3 auto-generated points. These points are randomly sampled from pixels where the mask prediction logits exceed a specific threshold (for Pascal VOC, we find that a threshold of 0.5 is generally sufficient). The output from the prompt encoder is then fed to the SAM head (i.e., mask decoder) along with the patch token outputs from the ViT backbone. Finally, the mask decoder produces fine-grained mask prediction logits for each class, and we designate the class with the highest logit value as the predicted class for each pixel.

## C WEIGHT AVERAGING

Weight averaging is a straightforward post-processing method proven to mitigate forgetting across a variety of fine-tuning tasks. Specifically, Wise-FT (Wortsman et al., 2022) proposes linearly in-

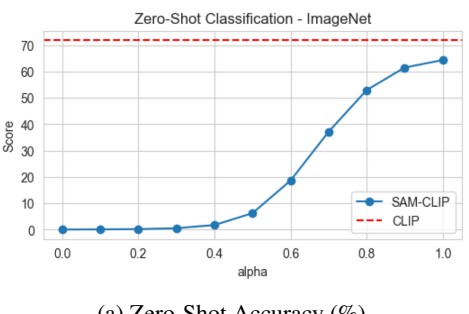
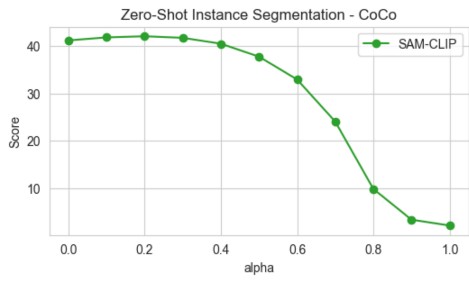

(a) Zero-Shot Accuracy (%)          (b) Zero-Shot Instance Segmentation (mAP)

Figure 7: Wise-FT (Wortsman et al., 2022) to a CLIP-distilled SAM ViT-B model. The red dashed line marks the performance of the CLIP teacher model.

terpolating the pretrained and fine-tuned parameters using a coefficient $\alpha$. In this study, we explore the application of Wise-FT in our setup. We focus exclusively on CLIP distillation applied to SAM ViT-B (serving as the student model), with a CLIP ViT-B/16 model acting as the teacher model. The model is trained on ImageNet-21k for 20 epochs. It is evident that the fine-tuned student model ($\alpha = 1$) gains zero-shot classification capabilities at the expense of forgetting its original zero-shot instance segmentation abilities. Upon applying Wise-FT to the fine-tuned model, we observe an inherent tradeoff between learning and forgetting. Notably, no optimal point exists where both high classification accuracy ($> 60\%$ on ImageNet) and a high mAP ($> 35$ mAP on COCO) are achieved simultaneously.

## D  LIMITATIONS

Our proposed method for merging existing foundational vision models may inherit the limitations of the original models. Specifically, our approach might carry over limitations from both the original SAM and CLIP models, including biases in data distribution. We have not assessed the robustness and fairness of our method in this work. Another potential limitation is the model size/architecture of the base VFM (SAM in this paper), which must be adopted from an existing model. However, we believe this should not be a practical limitation. The original SAM model offers several sizes/architectures (ViT-B/L/H). Moreover, follow-up works, such as MobileSAM (Zhang et al., 2023), could be adopted as the base model in our proposed method to achieve a suitable final merged model. Additionally, our merged image encoder for the auxiliary model (CLIP in this case) requires an additional head (the CLIP-Head here). In this work, this increases the overall size by approximately 25% compared to a single ViT-B.

## E  MORE DISCUSSIONS ON RELATED WORKS

Due to the page limit of the main text, we provide additional discussions of related works in this Appendix section.

**Composition of Separate SAM and CLIP Models**  It has been shown that composing SAM and CLIP for semantic segmentation is feasible by using SAM to generate all possible segmentation masks and then using CLIP to provide labels (IDEA Research, 2023). However, this approach requires loading two models simultaneously (2x memory footprint) and, for each image, needs one forward pass of the SAM backbone (under 1024 resolution) to generate $K$ object segments, followed by a forward pass of the CLIP model for each segment to filter (overall $K + 1$ passes). With SAM-CLIP , only one ViT model needs to be loaded (lower memory footprint), and a single forward pass of the ViT backbone is required for each image. Overall, our method offers significant efficiency advantages over the model composition approach in terms of memory and computational costs during inference.

