# OpenReview forum: "SAM-CLIP: Merging Vision Foundation Models towards Semantic and Spatial Understanding"
_ICLR.cc/2024/Conference — ICLR 2024 Conference Withdrawn Submission_

### Official Review · Reviewer_Unxf · 2023-10-29

**Soundness:** 2 fair
**Presentation:** 2 fair
**Contribution:** 3 good
**Rating:** 3
**Confidence:** 4

**Summary:**

This work aims to unify CLIP and SAM – two powerful vision foundation models (VFMs) – to enable a single set of parameters that are capable of retraining the advantages of both VFMs. The authors treat such model merging as a continual learning problem, where, given a pretrained VFM, the knowledge of a second VFM is merged without forgetting the initial knowledge.

The proposed model, SAM-CLIP, assumes access to a small part of pretraining data or its surrogates to be replayed during the merging process. The SAM model is used as the base VFM during the distillation, where CLIP is regarded as the auxiliary VFM and its knowledge is distilled via a cosine distillation loss. To avoid the catastrophic forgetting of SAM’s original capabilities, the authors propose a rehearsal-based multi-task distillation loss to gradually distill the external knowledge to the base VFM.

The resulting trained SAM-CLIP is able to perform zero-shot classification, image-text retrieval, instance segmentation, and semantic segmentation. Across several benchmark datasets, the authors show that SAM-CLIP can achieve state-of-the-art performance in a single-stage inference setup.

**Strengths:**

- The paper endeavors to create a comprehensive model by merging pre-trained vision foundation models, aligning with the contemporary trends in computer vision.
- The contributed SAM-CLIP stems from a continual learning perspective, which is intuitive. As a result, SAM-CLIP is capable of conducting multiple visual understanding tasks in a zero-shot manner.

**Weaknesses:**

- A glaring omission in the paper is the technical detail surrounding the cross-VFM distillation. A deeper dive into the methodology, choices of operations, and potential effects of the framework is necessary.
- The paper's structure and presentation could use refinement. The disproportionate emphasis on background and literature review, coupled with scant technical details, detracts from its overall coherence and depth.
- Benchmarking SAM-CLIP against prior models, particularly those based on SAM, would offer a more rounded perspective on its performance and advantages.

**Questions:**

- **Q1:** The efficiency of SAM-CLIP on edge devices is emphasized multiple times throughout the manuscript, particularly in the “Abstract” and “Introduction” sections. However, the empirical evidence supporting SAM-CLIP's performance on such devices seems absent. Could the authors elucidate the specifics of the claim regarding SAM-CLIP’s suitability for edge devices? The reviewer would like to know what the claim means by “apt for edge device applications”.

---

- **Q2:** When assessing zero-shot semantic segmentation, SAM-CLIP is exclusively juxtaposed with CLIP-based models. How does SAM-CLIP fare when contrasted with SAM-centric models, notably Semantic-SAM [R1] and SEEM [R2]?

---

- **Q3:** The “Proposed Approach” section might benefit from more detailed explanations regarding the design and implementation. In particular, how do you perform the joint training between head probing and multi-task distillation?  how is the balance between head probing and multi-task distillation maintained during joint training? What metrics or criteria guide the selection of appropriate stopping points for training?

---

- **Q4:** The “Background” section contains a profusion of general literature introductions. A more succinct and discerning review that delves into comparative analyses would greatly enhance its value.

---

- **Q5:** Notable typos appeared in the current illustration of this paper, which should be revised accordingly. For example:
  - Page 1, the last paragraph: there should be a space between “tasks” and “Fifty et al., 2021”.
  - Page 2, the first paragraph: “consuming massive amount …” should be “consuming a massive amount …”.
  - Page 2, the first paragraph: “how to access to …” should be how to access …”.
  - Page 2, the second paragraph: “generalization to diverse set of tasks” should be “generalization to diverse sets of tasks”.
  - Page 2, the third paragraph: “we allow access to small part of …” should be “we allow access to a small part of …”.
  - Page 3, the first paragraph: “With compromise of a negligible drop …” should be “With a compromise of a negligible drop …”.
  - Page 3, the second paragraph: “enable additional zero-shot capabilities” should be “enabled additional zero-shot capabilities”.
  - Page 3, the second paragraph: “on-top of …” should be “on top of …”.
  - Page 3, the third paragraph: “a model, and training recipe …” should be “a model, and a training recipe …”.
  - Page 3, the third paragraph: “and produce high-resolution segmentation mask” should be “and produces a high-resolution segmentation mask”
  - Page 3, the third paragraph: “but has not released …” should be “but have not released …”.
  - Page 3, the fourth paragraph: “They show transfer of the same …” should be “They show the transfer of the same …”.
  - Page 3, the fourth paragraph: “and demonstrate transfer of different zero-shot capabilities” should be “and demonstrate the transfer of different zero-shot capabilities”.
  - Page 3, the fourth paragraph: “as well as emergence of new zero-shot capability” should be “as well as the emergence of new zero-shot capability”.
  - Page 3, the fifth paragraph: “referring to loss of previously learned knowledge due to …” should be “referring to a loss of previously learned knowledge due to …”.
  - Page 4, the third paragraph: “to obtain segmentation mask” should be “to obtain segmentation masks”.
  - Page 4, the third paragraph: “and many forward passes, make their deployment …” should be “and many forward passes, making their deployment …”.
  - Page 4, the fourth paragraph: “the optimization algorithm is exploring the parameter space …” should be “the optimization algorithm explores the parameter space …”.
  - Page 5, the second paragraph: “and inherits its …” should be “and inherit its …”.
  - Page 5, the sixth paragraph: “which is the case of our experiment of …” should be “which is the case in our experiment of …”.

---

- **Q6:** How does SAM-CLIP perform under out-of-distribution or data corruption cases?

---

References

- [R1] F. Li, et al. “Semantic-SAM: Segment and Recognize Anything at Any Granularity.” arXiv preprint arXiv 2307.04767.

- [R2] X. Zou, et al. “Segment Everything Everywhere All at Once.” arXiv preprint arXiv  2304.06718.

**Details Of Ethics Concerns:**

No ethics concern or only a minor ethics concern is observed.

---

> ### Author Response · Authors · 2023-11-13
> **Author Response to Reviewer Unxf [Part 1]**
>
> We would like to thank the reviewer for their constructive feedback and comments. Below are our itemized responses. We hope that our responses address all questions and concerns and clarify our contributions.
>
> 1. [**Writing Suggestions**] We appreciate the reviewer's suggestions regarding the refinement of the writing and structure of the paper. These suggestions will be incorporated in the next revision.
>
> 2. [**Comparison with SAM-based Models**] Our primary goal with SAM-CLIP is to integrate the abilities of CLIP into the SAM model to achieve enhanced semantic understanding. We demonstrate that SAM-CLIP exhibits superior zero-shot semantic segmentation performance compared to previous methods. To our knowledge, as of the submission date of this paper, the state-of-the-art zero-shot semantic segmentation models were all CLIP-based; thus, we did not have SAM-based models for comparison in this task. It is important to note that the Semantic-SAM and SEEM models, as mentioned by the reviewer, are trained on image datasets with semantic segmentation annotations. Models trained with semantic segmentation annotations are not considered *zero-shot* in the literature. Therefore, these SAM-based models cannot be compared for the zero-shot semantic segmentation task. We thank the reviewer for bringing Semantic-SAM and SEEM to our attention. Relevant discussion will be included in the revision.
>
> 3. [**Suitability for Edge Devices**]
>    To run an application requiring both SAM and CLIP abilities, one would need to store and deploy both models on the device. However, by merging SAM and CLIP into a single ViT-B model, only one ViT model is required to complete tasks that previously needed two models. Moreover, for processing each image on the edge device, SAM-CLIP only requires one forward pass through the backbone, whereas using two separate models would need 2x forward passes (doubling the compute and memory footprint costs). Therefore, SAM-CLIP is advantageous in terms of storage, memory, and compute costs for edge devices.
>
>    Please note that we are not introducing a new architecture in this work, but proposing to merge existing ViTs into a single one. Hence, existing benchmarks on different devices are valid (e.g., the memory and runtime performance of SAM-CLIP on any device is identical to that of SAM with ViT-B on such a device).
>
> 4. [**More Details**]
>    We thank the reviewer for suggesting elaborating more on the training technical details and distillation background review. These points will be addressed in the next revision.
>    In general, all our design choices (mentioned in Sections 3, 4.1, and Appendix A) are based on the trade-off between forgetting SAM’s original capabilities and learning CLIP’s capabilities. Instance Segmentation (SAM’s capability), zero-shot classification (CLIP’s capability), and zero-shot semantic segmentation (a new joint capability) are the metrics we used to guide our choices.
>
>    Our training process consists of two stages: i) CLIP-head probing and ii) joint training of the SAM-head, CLIP-head, and the ViT backbone using a multi-task distillation loss. In the first stage, only the CLIP-head is trainable, and it is trained using a single CLIP distillation loss (cosine distance between embeddings). At this stage, all image batches come from only $D_{CLIP}$. This stage does not involve early stopping (we train for 20 epochs). The motivation for this step is to have a warm start for the CLIP-head in the next stage where we also allow modifying the backbone, similar to [1].
>
>    In the second stage, the SAM-head and the ViT backbone become trainable, and we have a multi-task objective: CLIP Distillation (Eq. 1) and SAM self-distillation (Eq. 2). The balance between the losses is determined by the coefficient $\lambda$, which we picked to optimize the aforementioned trade-off. Each training step for this stage is performed as follows:
>    - Pick a batch of 2048 images from $D_{CLIP}$, run the forward pass, and compute gradients backward from the CLIP Distillation loss (note that only parameters of the CLIP-head and ViT backbone will get gradients after this step).
>    - Pick a batch of 32 images from $D_{SAM}$, run the forward pass, and compute gradients backward from the SAM self-distillation loss (note that only parameters of the SAM-head and ViT backbone will get gradients after this step).
>    - Apply one optimization step (note that at this point, the parameters of the ViT backbone have accumulated gradients from the above two steps).
>
>    We early-stop after 16 epochs (out of a full training length of 20 epochs) as we observed more severe forgetting (as measured by instance segmentation performance on COCO) after the 16th epoch.

---

> > ### Author Response · Authors · 2023-11-13
> > **Author Response to Reviewer Unxf [Part 2]**
> >
> > 5. [**Out-of-Distribution Evaluation**] We evaluated SAM-CLIP on ImageNet-v2, a test set commonly used for out-of-distribution evaluation of image classifiers. The comparison with CLIP models is shown in Table 1. It can be observed that SAM-CLIP's performance is comparable to state-of-the-art CLIP models on ImageNet-v2. We further include zero-shot classification on the Places-365 dataset, a scene recognition dataset with a data distribution significantly different from that of ImageNet. We observe SAM-CLIP obtaining even slightly better performance compared to baseline CLIP models with a ViT-B backbone. We believe these results collectively imply the robustness of SAM-CLIP under distribution shifts.
> >
> >
> > ## References
> >
> > [1] Kumar, Ananya, et al. "Fine-tuning can distort pretrained features and underperform out-of-distribution." arXiv preprint arXiv:2202.10054 (2022).

---

### Official Review · Reviewer_a39d · 2023-10-30

**Soundness:** 3 good
**Presentation:** 3 good
**Contribution:** 2 fair
**Rating:** 5
**Confidence:** 5

**Summary:**

The paper proposes SAM-CLIP to build a unified model with both the strengths of SAM and CLIP.  SAM and CLIP is employed to share the same image encoder with two separate heads. Two phased are adopted during the KD process: 1) Head probing 2) Multi-task distillation. Also 40.8M images are used in the distillation process. The results are validated on zero-shot instance segmentation, semantic segmentation and classification benchmarks.

**Strengths:**

1. The paper has a good motivation on merging two visual foundation models, i.e., SAM and CLIP, into a unified model, such that the distilled model can obtain both semantic and spatial understanding.

2. The paper is well organized and easy to understand.

3. The experiments in Figure 1 and the experiment section show the distilled model retains both good zero-shot ability from SAM and CLIP.

**Weaknesses:**

1. When evaluating zero-shot semantic segmentation, as in Figure 3, the paper proposes a two-stage process to first using clip head for coarse masks predictions and taking it as input to SAM for refinement. Is the predicted masks by SegCLIP in Table 2 also refined by SAM? Can the authors also provide the zero-shot semantic segmentation without using geometric prompts?

2. When evaluating zero-shot instance segmentation, the performance decrease on LVIS is not negligible. This suggests that the ability of SAM is decreasing after the distillation process. Can the authors also provide comparison to HQ-SAM on zero-shot instance segmentation with the same bounding box as prompt? HQ-SAM [a] is also designed for minimal forgetting and efficient tuning for SAM but without discussion in related works or results comparison. Also, the paper misses MobileSAM in the related work section, which also uses knowledge distillation.

[a] Segment Anything in High Quality. NeurIPS, 2023.
[b] Faster Segment Anything: Towards Lightweight SAM for Mobile Applications. arXiv:2306.14289.

3. Since the paper mentions edge device applications in the abstract, what are the model size, speed and memory consumption of the proposed sam-clip comparing to SAM/CLIP?

4. What is the influence of the dataset scale in Merged-41M, for example reducing images by half or further increasing the image number? How does the paper decide the respective data percentage for CLIP and SAM training? Also, how to decide the distillation loss value scales for the sam head and clip head, like 1:10?

**Questions:**

Can the method deal with the instance segmentation not using bbox as prompt but using the semantics from CLIP? Overall I am positive about this paper and willing to raise scores if my concerns in the weakness can be well addressed.

---

> ### Author Response · Authors · 2023-11-13
> **Author Response to Reviewer a39d**
>
> We would like to thank the reviewer for their positive and constructive feedback. Below are our itemized responses. We hope that our responses address all questions and concerns and clarify our contributions.
>
> 1. [**Clarification on Semantic Segmentation Results**] The performance of SAM-CLIP reported in Table 2 is obtained using **only the CLIP-head**, **without** any SAM-head refinement -- for a fair comparison, the input images to SAM-CLIP, similar to those in the baseline methods (shown in Table 2), are resized to 448px.
>
>     Figure 3 illustrates that combining the CLIP-head and SAM-head can further enhance performance, although the results in Table 2 do not utilize the SAM-head. The performance of SAM-CLIP with SAM-head enhancement is showcased in Table 5. For example, on the Pascal-VOC dataset, SegCLIP achieves 52.6 mIoU, while our SAM-CLIP, using only the CLIP-head without any refinement, attains 60.6 mIoU (as shown in Table 2). If both CLIP and SAM heads are combined, our SAM-CLIP's performance can further improve to 66.0 mIoU (as shown in Table 5).
>
> 2. [**Difference from HQ-SAM**] SAM-CLIP's training integrates the abilities of CLIP into the SAM model, resulting in a multi-task zero-shot model. In contrast, HQ-SAM focuses on enhancing SAM's original ability and improves its instance segmentation by fine-tuning on high-quality instance segmentation datasets. It is important to note that SAM-CLIP has a completely different goal from HQ-SAM: we aim to make the SAM model multi-task, while HQ-SAM focuses on improving its original task. Consequently, HQ-SAM shows improvement in instance segmentation (e.g., COCO and LVIS), while ours experiences some performance drop, as our goal is not to enhance SAM's instance segmentation ability. We thank the reviewer for bringing the HQ-SAM and MobileSAM papers to our attention. We will include the related discussion in the next revision.
>
> 3. [**Model Size**] We use the SAM ViT-B version as our base model architecture. After the merging process, a single ViT-B image encoder of SAM-CLIP can serve to perform original SAM tasks (prompt-based segmentation), CLIP tasks (prompt-based classification and retrieval), and new tasks of text-to-segmentation (i.e., zero-shot/text-prompted semantic segmentation). Note that all these tasks can be done with a single inference of ViT-B and require loading only one image encoder. We should note that for CLIP-related tasks, SAM-CLIP has an additional light-head (3 transformer layers) which adds 25% to the memory footprint compared to a stand-alone CLIP with ViT-B (12 transformer layers). Thanks for the suggestion. We will clarify this in the next revision.
>
> 4. [**Dataset Choice and Loss Coefficient**] In selecting the dataset, we primarily followed the approach of EVA-CLIP[1], which also distilled a CLIP model, using CC3M, CC12M, and ImageNet-21k. We additionally included YFCC-15M, a subset of the 400M data used to train the original OpenAI’s CLIP model.
>
>    The loss coefficient ratio of 1:10 for the CLIP and SAM heads was empirically determined from three options: 1:1, 1:10, 1:100. We found that 1:10 offers the best trade-off between mitigating the forgetting of SAM’s ability and learning CLIP’s ability. We will clarify this process in the next revision.
>
> 5. [**Instance Segmentation Without Bounding Box as Prompt**] If we understand your question correctly, you are referring to zero-shot semantic segmentation. This task involves providing an image and a set of candidate object class names (e.g., “dog”, “cat”, “human”) and asking the model to perform semantic segmentation. We have provided quantitative results for this task in Table 2. Please let us know if we have misunderstood your point.
>
> ## References
>
> [1] Fang, Yuxin, et al. "EVA: Exploring the limits of masked visual representation learning at scale." Proceedings of the IEEE/CVF Conference on Computer Vision and Pattern Recognition. 2023.

---

### Official Review · Reviewer_K2ur · 2023-10-31

**Soundness:** 3 good
**Presentation:** 3 good
**Contribution:** 2 fair
**Rating:** 5
**Confidence:** 4

**Summary:**

This paper merges CLIP and SAM, the two foundation models, into a single one that assimilates both knowledge and expertise learned separately. Specifically, the technical contributions include a reasonable finetuning design and integration of the two distillation losses. The resulting model supports language-driven prompts and enjoys a high-quality segmentation result.

**Strengths:**

1. This paper presents a simple yet effective way to merge two foundation models into a single one, and it inherits both advantages and demonstrates a significant performance boost;
2. The paper is well-organized, clearly written, and easy to follow;
3. The resulting model is promising and helpful for future research.

**Weaknesses:**

1. The resulting model inherits the zero-shot capability of CLIP, as demonstrated in Table 1-5. However, it seems that there is no evidence showing the resulting model does not suffer from catastrophic forgetting. Even though the segmentation performance is better than CLIP-head prediction, it still doesn't compare with the segmentation result of SAM and it is unclear how much performance is degraded compared to the original SAM. The demo in Figure 3 shows that the SAM-head refined output is still filled with some artifacts and seems to have a large performance gap with the original SAM.
2. The proposed method is limited to the sizes of released SAM models. Since the vision encoder must be initialized SAM vision encoder, we cannot obtain a resulting model with an arbitrary size.

**Questions:**

1. The authors should explain more clearly the performance gap with the original SAM in terms of segmentation quality.
2. The authors should also give the output of the original SAM, with the same examples shown in Figure 3.
3. The authors should discuss more limitations with the resulting model and the proposed method.

If the above concerns are addressed, I am willing to improve the rating.

---

> ### Author Response · Authors · 2023-11-13
> **Author Response to Reviewer K2ur**
>
> We would like to thank the reviewer for their positive and constructive feedback. Below are our itemized responses. We hope that our responses address all questions and concerns and clarify our contributions.
>
> 1. [**Clarification on Catastrophic Forgetting**] We have conducted studies on the catastrophic forgetting of SAM-CLIP versus SAM, as shown in Table 1. The last two columns of Table 1 for COCO and LVIS demonstrate that SAM-CLIP experiences some mild performance degradation compared to SAM in instance segmentation. This indicates that while SAM-CLIP does exhibit some degree of forgetting, it is not catastrophic. To provide a more intuitive understanding of the segmentation quality of SAM versus SAM-CLIP, we present two image examples and their corresponding segmentation predictions by SAM and SAM-CLIP in Appendix A.1 of the newly updated manuscript. Please check out the current rebuttal revision of the paper [[PDF](https://openreview.net/pdf?id=GKau1ekOtH)].
>
> 2. [**Model Size**] SAM releases three ViT versions of varying sizes: ViT-B(ase), ViT-L(arge), and ViT-H(uge), covering most ViT use cases. In the paper, our experiments were conducted only with ViT-B, but this method can be directly applied to ViT-L and ViT-H as well. The proposed approach can also be applied to subsequent works of SAM, such as MobileSAM [1], to obtain other model sizes for SAM-CLIP.
>
> 3. [**Performance Gap with the Original SAM**] For a quantitative comparison with the original SAM, we have conducted _instance segmentation_ studies. We acknowledge that SAM-CLIP has a small performance gap compared to SAM in terms of instance segmentation ability, as shown in the last two columns of Table 1: (-0.3% and -1.8% mAP on COCO and LVIS datasets, respectively). However, compared to SAM, our SAM-CLIP has broader zero-shot capabilities (Fig 1) and representation quality (Fig 4). Furthermore, in the newly [revised manuscript](https://openreview.net/pdf?id=GKau1ekOtH), we provide two image examples in Fig. 5 of Appendix A.1 to qualitatively show that the instance segmentation outputs of SAM and SAM-CLIP have similar quality.
>
> 4. [**Qualitative Comparison of Segmentation**] Thank you for the suggestion on qualitative segmentation comparison. In the newly uploaded [revision](https://openreview.net/pdf?id=GKau1ekOtH), we provide segmentation outputs of the original SAM given the same point prompts. One can see from Fig. 6 in Appendix A.1 that SAM tends to segment only a sub-part of the object class pinpointed by the point prompts, instead of segmenting the whole semantic object class. This visual comparison implies that the combination of CLIP and SAM abilities in our SAM-CLIP is quite crucial for (zero-shot) semantic segmentation.
>
> 5. [**Discussion of More Limitations**] We will further discuss additional limitations of our trained model and the proposed method in the next revision, including the above discussion on model size.
>
> ## References
>
> [1] Zhang, Chaoning, et al. "Faster Segment Anything: Towards Lightweight SAM for Mobile Applications." arXiv preprint arXiv:2306.14289 (2023).

---

### Official Review · Reviewer_Kk2t · 2023-11-02

**Soundness:** 2 fair
**Presentation:** 3 good
**Contribution:** 2 fair
**Rating:** 5
**Confidence:** 5

**Summary:**

This paper proposes a distillation paradigm to incorporate SAM and CLIP, combing their instance segmentation and semantic recognition capabilities. SAM-CLIP uses extensive pre-training data from original models and learns a unified encoder along with two task-specific heads. SAM-CLIP showcases good performance across zero-shot classification and segmentation tasks.

**Strengths:**

1. The motivation is reasonable to combine SAM and CLIP to infuse their own advantages.

2. SAM-CLIP shows good performance on zero-shot semantic segmentation tasks.

3. The writing is clear and easy to follow.

**Weaknesses:**

1. The contribution is a little overclaimed as *'we introduce a simple recipe to efficiently merge VFMs into a unified model that assimilates their expertise.'*. I think this method is specifically designed for CLIP and SAM, and cannot be simply generalized to other VFMs.

2. The cost of training SAM-CLIP is expensive. The training data includes many sources up to 41M. Considering CLIP and SAM have already cost large-scale pre-training resources, continually tuning them as SAM-CLIP is not cost-effective. Although SAM-CLIP achieves good results for semantic segmentation, it hurts the original performance of both SAM and CLIP. I think simply cascading SAM and CLIP in a training-free way (CLIP generates prompt by vision-language alignment and then SAM segments or SAM segments all objects and then CLIP classifies) can obtain even comparable results to SAM-CLIP, which is more practical in real-world applications.

**Questions:**

SAM itself can also be prompted by texts (semantics), though not open-sourced. What's the advantage of SAM-CLIP compared to SAM with text prompt?

---

> ### Author Response · Authors · 2023-11-13
> **Author Response to Reviewer Kk2t**
>
> We would like to thank the reviewer for their constructive feedback and comments. Below, please find our itemized responses. We hope that our responses address all questions and concerns and clarify our contributions.
>
> 1. [**Merging VFMs Other Than CLIP and SAM**] Our proposed merging technique, based on an embedding distillation loss and an unlabeled image dataset, can be utilized beyond CLIP and SAM with any image encoder. We will clarify in the next revision how one might use the same approach to merge other vision encoders.
>
>     For instance, one can merge DINOv2[1] with SAM (or SAM-CLIP) in the same pipeline, by substituting the CLIP teacher model with a DINOv2 model. In this paper, we focus on SAM and CLIP because they are both zero-shot models with different capabilities, and merging them provides a vision model with broader zero-shot abilities. In contrast, DINOv2 is a pre-trained image backbone (with strong representation learning ability) that does not operate in a zero-shot manner. However, merging with DINOv2 could enhance representation learning and potentially offer some benefits. For example, one can also merge CLIP and DINOv2 using our pipeline (e.g., using CLIP as the base model and then applying multi-task distillation with CLIP and DINOv2 as teacher models), and possibly obtain a merged model with stronger representation suitable for certain downstream tasks.
>
> 2. [**Training and Runtime Cost of SAM-CLIP**]
>    We believe there is a misunderstanding regarding the training cost of SAM-CLIP.
>    SAM-CLIP is trained on a 41M image dataset, which is *20-30x smaller* than the training sets of SAM and CLIP. For instance, SAM is trained on SA-1B, and we use only a 5.7% subset from SA-1B for the SAM self-distillation of our model; the state-of-the-art CLIP model is trained on DataComp-1B (1.4B image-text pairs), and the data we use for CLIP distillation (40M unlabeled images) represents less than 3% of DataComp-1B. Hence, the training cost of SAM-CLIP is quite small compared with the training of SAM and CLIP.
>
>    Regarding deployment, it has been shown that composing SAM and CLIP for semantic segmentation is feasible by using SAM to generate all possible segmentation masks and then using CLIP to provide labels [3]. However, this approach requires loading two models simultaneously (*2x memory footprint*) and, for each image, needs a forward pass of the SAM backbone (under 1024 resolution) to generate K object segments, followed by a forward pass of the CLIP model for each segment to filter (overall K+1 passes).
>
>    With SAM-CLIP, only one ViT model needs to be loaded (lower memory footprint), and a single forward pass of the ViT backbone is required for each image. Overall, our method offers significant efficiency advantages over the model composition approach in terms of memory and computational costs during inference.
>
>    In the next revision, we will elaborate further on the computational costs of existing training-free approaches for zero-shot semantic segmentation.
>
> 3. [**Comparison with SAM**]
>    We cannot comment on the performance of the SAM model trained with text (using a different pre-training loss than the public SAM) as it is unreleased. The SAM paper [2] refers to it as a “Preliminary Exploration” and does not provide any quantitative results (such as for zero-shot semantic segmentation as we do).
>
>    Regarding capabilities, SAM-CLIP can perform all zero-shot tasks of CLIP (zero-shot classification & image-text retrieval), which SAM (both the public and unreleased versions) cannot. Furthermore, we want to emphasize that our motivation is not solely to improve the SAM or CLIP model but to demonstrate the merging of two foundational models with different zero-shot capabilities into one, with SAM+CLIP serving as a demonstrative case.
>
> ### References
>
> [1] Oquab, Maxime, et al. "Dinov2: Learning robust visual features without supervision." arXiv preprint arXiv:2304.07193 (2023).
>
> [2] Kirillov, Alexander, et al. "Segment anything." arXiv preprint arXiv:2304.02643 (2023).
>
> [3] "Grounded Segment-Anything." https://github.com/IDEA-Research/Grounded-Segment-Anything (2023)

---

### Author Response · Authors · 2023-11-13
**General Response to All Reviewers & 1st Rebuttal Revision**

We would like to thank all reviewers for the time they have dedicated to providing constructive comments on our work. We are pleased with the positive feedback we received from all reviewers: the strong motivation to merge vision foundation models (Kk2t and A39d), which aligns with contemporary trends in computer vision (Unxf), the simple yet effective method of merging (K2ur), and the significant boost in performance (K2ur, Kk2t). The SAM-CLIP model is seen as promising and helpful for future research (K2ur), retaining both the robust zero-shot ability from SAM and CLIP (a39d), and capable of conducting multiple visual understanding tasks in a zero-shot manner (Unxf). We are also glad that reviewers (Kk2t, K2ur, a39d) find the paper's presentation clear, well-organized, and easy to understand.

## Rebuttal Revision Version

We have revised the paper and uploaded this rebuttal revision version. Please review it at [this link](https://openreview.net/pdf?id=GKau1ekOtH). Notably, we added **Appendix A.1**, which provides qualitative (visual) comparisons between SAM and our SAM-CLIP for instance segmentation and semantic segmentation, in Figs. 5 and 6, respectively.

[**Instance Segmentation Comparison**] Specifically, in Fig. 5, we use two image examples to demonstrate that the instance segmentation outputs of SAM and SAM-CLIP are quite similar, reflecting the comparable quantitative instance segmentation performance for the two methods shown in Table 1.

[**Semantic Segmentation Comparison**] In Fig. 6, we use two image examples (the same ones used in Fig. 3) to visually compare the semantic segmentation masks of SAM-CLIP and SAM, as requested by Reviewer K2ur. We feed point prompts (generated by the CLIP-head of SAM-CLIP) to SAM and show that its predicted segmentation masks typically only segment a sub-part of the object class indicated by the point prompts. This demonstrates that the point prompting approach of SAM is insufficient for semantic segmentation, as SAM does not understand the semantic meanings of object classes (e.g., “dog,” “human,” “horse”). In contrast, our SAM-CLIP combines the segmentation ability of SAM with the semantic understanding of CLIP, enabling strong zero-shot semantic segmentation ability.

---

### Author Response · Authors · 2023-11-14
**2nd Rebuttal Revision**

Dear reviewers

We have revised the paper again and uploaded the second version of the rebuttal revision. Please review it at [this link](https://openreview.net/pdf?id=GKau1ekOtH). Specifically, we have included the following changes to the manuscript, addressing a collection of comments from all reviewers (Kk2t, K2ur, a39d, Unxf):

+ [a39d, Unxf] Added a new paragraph in Sec. 2, discussing recent SAM-based works including HQ-SAM, MobileSAM, FastSAM, Semantic-SAM, and SEEM.

+ [K2ur, a39d] Enhanced the discussion of limitations regarding the model size and architecture of our SAM-LIP in Appendix D (titled “Limitations”).

+ [Unxf] Included a paragraph on training details for the multi-task distillation stage of SAM-CLIP in Appendix A.

+ [a39d] Added a paragraph in Appendix A detailing how the loss coefficients for SAM-distillation and CLIP-distillation losses were empirically determined.

+ [Kk2t] Introduced a new section, Appendix E, to discuss the training-free approach of composing separate SAM and CLIP models.

+ [Unxf] Addressed typos mentioned by Reviewer Unxf.

We kindly request your valuable feedback on our response and the revised manuscript to ensure that we have adequately addressed your concerns and comments.

---

### Author Response · Authors · 2023-11-17
**Invitation to Engage in Author-Reviewer Discussion on Paper "SAM-CLIP"**

Dear Reviewers,

We want to thank you again for providing your valuable feedback on our paper, "SAM-CLIP: Merging Vision Foundation Models towards Semantic and Spatial Understanding."

We have addressed all the reviewer comments in our rebuttal and author responses, which were submitted 4 days ago. We are writing to kindly request you to take a look at our detailed responses when you get a chance. We would greatly appreciate any additional feedback you may have or if you could help engage in a discussion with us.

As authors, we are very eager to address any remaining concerns you may have and clarify any parts of our work to strengthen the paper. Please let us know if you need any additional information from our side.

Your expertise and perspective would be invaluable during this discussion process. We hope to learn from you and improve the work through this open review.

Sincerely,
Authors of Submission 6056